# The Influence of Parental Knowledge and Basic Psychological Needs Satisfaction on Peer Victimization and Internet Gaming Disorder among Chinese Adolescents: A Mediated Moderation Model

**DOI:** 10.3390/ijerph18052397

**Published:** 2021-03-01

**Authors:** Qiao Liang, Chengfu Yu, Qiang Xing, Qingqi Liu, Pei Chen

**Affiliations:** Department of Psychology and Research Center of Adolescent Psychology and Behavior, School of Education, Guangzhou University, Guangzhou 510006, China; 2111808166@e.gzhu.edu.cn (Q.L.); liuqingqi@mails.ccnu.edu.cn (Q.L.); a13430205519@163.com (P.C.)

**Keywords:** peer victimization, internet gaming disorder (IGD), basic psychological needs satisfaction (BPNS), parental knowledge

## Abstract

Ample evidence indicates that peer victimization is a crucial risk factor for adolescent internet gaming disorder (IGD); however, little is known about the mechanisms underlying this association. Based on the risk-buffering model and self-determination theory, this study tested whether parental knowledge moderated the relationship between peer victimization and adolescent IGD and whether this moderating effect was mediated by basic psychological needs satisfaction (BPNS). A sample of 3080 adolescents (*Mean*_age_ = 14.51; *SD* = 1.97) anonymously responded to a set of questionnaires. The results revealed that the positive association between peer victimization and adolescent IGD was stronger among adolescents with parents who had low-level parental knowledge than for those with high-level parental knowledge. Moreover, this moderating effect was mediated by BPNS. These findings highlight that parental knowledge is an important protective factor against IGD for adolescents who experience peer victimization and BPNS is one mechanism that explains how this effect works.

## 1. Introduction

Internet gaming disorder (IGD) refers to the physical, psychological, and social damage caused by the uncontrollable, excessive, and compulsive playing of internet games [1,2]. In March 2020, a survey in China indicated that the number of internet users aged 10–19 years had reached 174 million, and that the rate of playing online games had reached 58.9% among the total number of internet users [3]. Recent studies report that Chinese adolescents display a high prevalence of IGD, ranging within 2.97–13% [4,5,6]. As China has numerous gamers and a high prevalence of IGD among adolescents, the causes, correlation, and consequences of IGD among Chinese adolescents have garnered research attention [6,7,8]. However, the research in this field is still in its infancy, and the identification of factors and potential mechanisms that influence the development of IGD is needed to develop effective evidence-based interventions to prevent IGD among adolescents.

Adolescence is an important period for building friendships and seeking peer acceptance [9]. Negative peer relationships (especially peer victimization) may be an adverse stressor that contributes to the development of problem behaviors among adolescents [10]. Peer victimization is defined as experiences of any form of attack from a peer, including verbal, physical, and relational aggression [11]. As a stressor, peer victimization can contribute to the occurrence of painful and humiliating feelings, as well as psychological distress and psychological maladjustment [12,13]. Experiencing peer victimization can lead to adolescents adopting maladaptive coping strategies, such as relying on online gaming to diminish psychological distress and relieve negative emotions [14]. When adolescents suffer from IGD as a result of excessive online gaming, they are more likely to experience several negative consequences, including mental health problems, social difficulties, a poor academic performance, and physical harm [15,16]. The positive relationship between peer victimization and adolescent IGD has been consistently supported by several empirical studies [17,18,19]. For instance, among a sample of 1401 Chinese middle school students, Chen et al. [17] observed that adolescents who experienced peer victimization were at an increased likelihood of developing IGD. This body of research demonstrates that peer victimization is a particularly critical risk factor for adolescent IGD.

Although the positive link between peer victimization and adolescent IGD has received empirical support, research examining the mechanisms underlying this relationship remains limited. Revealing the mediating and moderating mechanisms of the association between peer victimization and adolescent IGD can provide meaningful information for interventions. Therefore, based on the risk-buffering model [20,21] and self-determination theory [22,23], this study aimed to explore whether parental knowledge moderates the relationship between peer victimization and IGD, and whether this moderating effect is mediated by basic psychological needs satisfaction.

### 1.1. Parental Knowledge as a Moderator

While a risky environment can increase the likelihood of developing problematic behaviors, adolescents can demonstrate positive adjustments and resilient outcomes when influenced by positive factors. According to the risk-buffering model [20,21], protective factors are particularly beneficial in neutralizing the negative impact of risks. When protective factors are present, the negative effects of adversity on adolescent development can be effectively reduced; however, if such protective factors do not exist or cease to exist, the predictive effect of risk factors on negative development will be significantly enhanced [20,21]. In other words, the detrimental effect of peer victimization on IGD might be attenuated by a protective factor. Notably, prior studies found that parental knowledge was a robust protective factor that helps to buffer the effect of a risky environment and reduce the emergence of deviant behaviors [17,24,25]. For example, both Jiang et al. [24] and Chen et al. [17] documented that parental knowledge plays a meaningful role in decreasing the impact of peer victimization. Likewise, Lahey et al. [25] indicated that adolescents showed less delinquency in late adolescence when they acquired parental knowledge in early adolescence. Based on the studies and theories mentioned above, parental knowledge may be considered a protective factor that might prevent the risk of IGD when an adolescent experiences peer victimization.

Parental knowledge refers to the extent to which parents are aware of their adolescents’ daily experiences, including their whereabouts, activities, and companions [26,27]. Parental knowledge regarding one’s adolescent child results from positive parent–child communication, which includes parents actively asking about, and adolescents willing to share, their life experiences [26]. Unlike tracking or surveillance, parental knowledge develops in a close and trusting parent–child relationship, which may be the best process by which to diminish adolescents’ problem behaviors [26,27,28]. Existing empirical evidence demonstrates that adolescents who have parents with a greater degree of parental knowledge may experience greater resilience against adversity, such as peer victimization [24,29]. Moreover, parental knowledge may reduce the risk of the occurrence of behavioral problems, including IGD [30,31,32,33]. More importantly, parental knowledge could significantly moderate the relationship between peer victimization and addictive behaviors among adolescents [24,33]. For instance, Jiang et al. [24] reported that a high level of parental knowledge buffered the negative effect of peer victimization on adolescent substance use. Similarly, Tian et al. [33] demonstrated that the association between deviant peer affiliation and IGD was attenuated for adolescents who reported greater parental knowledge. Overall, it appears that a high level of parental knowledge can attenuate the adverse effects of peer victimization on adolescent IGD. Therefore, we hypothesized the following:
**Hypothesis** **1** **(H1).**Parental knowledge moderates the relationship between peer victimization and adolescent IGD. Specifically, the positive link between peer victimization and IGD is stronger for adolescents with low levels of parental knowledge than for those with high levels of parental knowledge.

### 1.2. Basic Psychological Needs Satisfaction as a Mediator

Basic psychological needs satisfaction (BPNS) is an essential condition for healthy psychological development. According to self-determination theory [22,23], BPNS is composed of autonomy (i.e., the feeling that you can determine your behavior), relatedness (i.e., building connections with others), and competence (i.e., handling problems with your own abilities). An individual’s motivation, behaviors, and sense of well-being are decided by the degrees to which these three needs are satisfied [22]. A favorable environment can satisfy an individual’s psychological needs and thus contribute to positive psychological development. However, an unfavorable environment (e.g., one characterized by peer victimization) can inhibit the satisfaction of basic psychological needs [23,34]. Therefore, an individual might develop compensatory mechanisms to assist them in satisfying their needs in particular contexts [34]. It is worth noting that online games are often considered the “best place” to seek psychological compensation among adolescents who experience frustrated psychological needs. As online games can have a powerful effect on satisfying adolescents’ psychological needs [35,36], they provide a strong motivation for continuous engagement in online gaming, which may eventually lead to IGD [35,37].

BPNS can be impaired by peer victimization [38,39], given that, when adolescents suffer from peer victimization, they cannot establish intimate relationships with their peers, and their development of competence and control over their lives is damaged. Therefore, adolescents’ needs for relatedness, autonomy, and competence are unfulfilled or frustrated. Furthermore, the occurrence of IGD may be exacerbated due to the frustration of not having one’s basic psychological needs satisfied. Some researchers have suggested that seeking psychological needs satisfaction from online gaming may increase when adolescents’ basic psychological needs decrease, as online games provide a common means for youth to obtain psychological satisfaction [35,37,40]. Previous research indicates that BPNS mediates the relationship between negative circumstances (e.g., stressful life events and peer victimization) and adverse outcomes (e.g., internet addictive behaviors and anxiety) [39,41]. Therefore, BPNS may play a mediating role in the association between peer victimization and IGD.

BPNS could function as an important psychological process to explain the moderating effect of parental knowledge in the relationship between peer victimization and adolescent IGD. For instance, compared with low-level parental knowledge, adolescents with parents with a high level of parental knowledge were less likely to experience the negative effects of peer victimization [24,29] and seldom experienced mental health problems [42]. Similarly, Ang et al. [43] indicated that as parents’ knowledge increased, the link between loneliness (i.e., the representation of inadequate basic psychological needs satisfaction) and problematic internet use was weakened. According to the research mentioned above, we speculated that BPNS could be a possible mediator of the moderating influence of parental knowledge on the relationship between peer victimization and adolescent IGD. Consequently, we proposed the following hypothesis:
**Hypothesis** **2** **(H2).**BPNS mediates the moderating effect of parental knowledge on the association between peer victimization and adolescent IGD.

### 1.3. The Present Study

As the occurrence of IGD in adolescence might be influenced by factors across domains (including the family, peers, and individuals), factors from the single domain are difficult to address and prevent IGD among adolescents. The findings mentioned above suggest that peer victimization may be a salient predictor of IGD and that parental knowledge may play an important role in buffering the association between peer victimization and IGD. Moreover, this moderating effect could be explained by BPNS. The present study’s exploration of these factors contributes to the construction of a meaningful intervention plan for IGD, which is a matter that has seldom been given attention in the existing literature—to the best of our knowledge, no previous research has explored the combined influences of peer victimization, parental knowledge, and the satisfaction of basic psychological needs on IGD among adolescents. Based on the risk-buffering model [20,21] and self-determination theory [22,23], this study constructed a mediated moderation model to fill this gap. Specifically, the present study aimed to test (1) whether parental knowledge moderated the relationship between peer victimization and IGD and (2) whether BPNS mediated this moderating effect. Figure 1 illustrates the proposed research model.

## 2. Methods

### 2.1. Participants

Participants consisted of 3080 adolescents (male = 47%, *n* = 1447) from 10 public secondary schools in Guangdong Province in southern China, using random cluster sampling techniques. The ages of the adolescents ranged from 10 to 19 years (*Mean_age_* = 14.51 years, *SD* = 1.97 years). Approximately 43.5% of the adolescents were from the middle school and 53.5% were from the senior school.

### 2.2. Measures

#### 2.2.1. Peer Victimization

Peer victimization was measured using the Chinese version of the Peer Victimization Questionnaire [44]. Adolescents were asked to report how often they experienced physical, relation, and verbal victimization from their peers during the past six months. All items were rated on a five-point Likert scale (1 = never to 5 = four or more times). The peer victimization score was calculated by averaging all items, with higher scores reflecting higher levels of peer victimization. The Cronbach’s alpha was 0.86 for this questionnaire in this study. In Chinese adolescents, this questionnaire demonstrated good validity and reliability [29,45].

#### 2.2.2. IGD

IGD was assessed using the Chinese version of the IGD Questionnaire [46]. Adolescents were required to report the frequency of their experienced symptoms of IGD in the past six months. All items were rated on a three-point Likert scale: 0 = never; 0.5 = sometimes; and 1 = yes. The scale was scored by calculating the average of the 11 items, with higher scores indicating greater severity of IGD. The Cronbach’s alpha was 0.80 for this questionnaire in this study. The questionnaire’s validity and reliability were affirmed in previous studies [47,48].

#### 2.2.3. BPNS

The Chinese version of the BPNS Questionnaire [49] was used to assess the adolescents’ BPNS. The original version of the scale was developed by Gagné [50]. Adolescents were required to report the extent of their psychological needs satisfaction in the past six months. This questionnaire consisted of three subscales, including Autonomy Need (seven items, e.g., “I am free to do something and live my life in my own way”), Relatedness Need (eight items, e.g., “It’s easy for me to make friends”), and Competence Need (six items, e.g., “I have the ability to deal with my problems”). All items were rated on a five-point Likert scale from 1 = not at all true to 5 = very true. The scores for the three subscales were summed to create a total score, with higher scores indicating higher levels of BPNS. The Cronbach’s alpha was 0.81 for this scale in this study. This questionnaire has also been found to have good validity and reliability in a group of Chinese adolescents [51].

#### 2.2.4. Parental Knowledge

Parental knowledge was measured by using the five-item Chinese version of the parental knowledge questionnaire [24]. Participants were asked to report the degree to which their parents were aware of their activities and companions during the past six months. A sample item included “Do your parents know what you do in your leisure time?” All items were rated on a three-point Likert scale from 1 = knows little to 3 = knows much. The average score of all items was calculated, with higher scores indicating higher levels of parental knowledge. The Cronbach’s alpha was 0.77 for this questionnaire in this study. This questionnaire has been demonstrated to have good validity and reliability in previous research [29,33].

#### 2.2.5. Control Variables

The participant’s gender, age, impulsivity, and parent–adolescent relationship were included as covariates as they are correlated with IGD [8,52,53]. Impulsivity was measured using the Chinese Version of the UPPS-P impulsive behavior scale [54]. The parent–adolescent relationship was measured using the Chinese version of the Parent–Adolescent Relationship Questionnaire [55]. In this study, the Cronbach’s alphas of the UPPS-P Scale and Parent–Adolescent Relationship Questionnaire were 0.77 and 0.80, respectively.

### 2.3. Procedure

Written informed consent was obtained from the adolescents, their parents, and the school administrator before data collection began. The questionnaires were administered to the students during class time by well-trained researchers. The research assistants informed the adolescents that their participation was voluntary and their questionnaire responses would be confidential and anonymous. As such, they were asked to respond honestly to all questionnaire items and could omit any question that made them feel uncomfortable. Adolescents had 30 min to complete the questionnaire. Moreover, those who completed the questionnaire received stationery as a token of our appreciation. The survey materials and study procedures were approved by the Ethics in Human Research Committee of the Department of Psychology, Guangzhou University (protocol number: GZHU2019012; date of approval: 2019/05/27).

### 2.4. Statistical Analyses

Descriptive statistics were computed using SPSS 20.0 (IBM, Armonk, NY, USA). Mplus 7.1 (Muthén & Muthén, Los Angeles, CA, USA) was used for the structural equation modeling to test the mediating and moderating effects [56]. The present study adopted the mean imputation to address missing data [57]. An acceptable model fit was assessed by three indices: χ^2^/*df* value ≤ 5; a comparative fit index (CFI) ≥ 0.90; and a root mean square error of approximation (RMSEA) ≤ 0.06 [58]. In addition, we used a bootstrapping procedure to test the statistical significance of the indirect effects [59]. We interpreted the indirect effects by computing the 95% bias-corrected bootstrapped confidence intervals (CIs) with 1000 resamples. CIs excluding zero indicated significance at α = 0.05.

## 3. Results

### 3.1. Preliminary Analyses

According to the diagnostic criterion for IGD [2], almost 4.25% (*n* = 131) of the adolescents in the present sample were classified as IGD; this proportion is in line with national Chinese adolescent data [60] and recent studies [46,48]. Notably, approximately 0.5% of data were missing in the present study. Table 1 presents the means, standard deviations, and correlation coefficients for all variables. As expected, the results revealed that peer victimization positively correlated with IGD (r = 0.22), and both parental knowledge (r = −0.25) and BPNS (r = −0.22) negatively correlated with IGD. In addition, peer victimization negatively correlated with BPNS (r = −0.37).

### 3.2. The Moderating Effect of Parental Knowledge on the Direct Relationship between Peer Victimization and Adolescent IGD

The moderated model presented in Figure 2 was found to be an acceptable fit for the data: χ^2^/*df* = 3.36; CFI = 0.96; and RMSEA = 0.06. The results demonstrated that the main effects of peer victimization (*β* = 0.11, *SE* = 0.02, *t* = 6.48, *p* < 0.01, 95% CI [0.076, 0.142]) and parental knowledge (*β* = −0.14, *SE* = 0.02, *t* = −7.83, *p* < 0.01, 95% CI [−0.175, −0.105]) on IGD were significant. Moreover, the interaction effect between peer victimization and parental knowledge on IGD was significant (*β* = −0.05, *SE* = 0.02, *t* = −3.11, *p* < 0.01, 95% CI [−0.081, −0.018]).

We conducted a test of simple slopes to further explore the significant interaction of peer victimization × parental knowledge (Figure 3). The positive relationship between peer victimization and IGD was stronger among adolescents who reported lower levels of parental knowledge (1 *SD* below *M*; *β* = 0.16, *SE* = 0.02, *t* = 7.31, *p* < 0.01, 95% CI [0.116, 0.202]) than for those who reported higher levels of parental knowledge (1 *SD* above *M*; *β* = 0.06, *SE* = 0.02, *t* = 2.42, *p* < 0.05, 95% CI [0.011, 0.108]).

### 3.3. The Moderating Effect of Parental Knowledge on the Indirect Relationship between Peer Victimization and IGD

The mediated moderation model (Figure 4) demonstrated a good fit to the data: χ^2^/*df* = 2.82; CFI = 0.97; and RMSEA = 0.05. The interaction between peer victimization and parental knowledge on BPNS was significant (*β* = −0.04, *SE* = 0.01, *t* = −2.79, *p* < 0.01, 95% CI [−0.067, −0.012]). Figure 5 presents the predicted BPNS as a function of peer victimization and parental knowledge. Specifically, the negative relationship between peer victimization and BPNS was significantly stronger among adolescents who reported high levels of parental knowledge (1 *SD* above *M*; *β* = −0.28, *SE* = 0.02, *t* = -12.83, *p* < 0.01, 95% CI [−0.324, −0.238]) than for adolescents who reported low levels of parental knowledge (1 *SD* below *M*; *β* = −0.20, *SE* = 0.02, *t* = −10.43, *p* < 0.01, 95% CI [−0.239, −0.164]). Therefore, the relationship between peer victimization and BPNS was moderated by parental knowledge.

Moreover, BPNS was negatively related to IGD (*β* = −0.09, *SE* = 0.02, *t* = −4.36, *p* < 0.01, 95% CI [−0.128, −0.049]), and this relationship was also moderated by parental knowledge (*β* = 0.04, *SE* = 0.02, *t* = 2.48, *p* < 0.05, 95% CI [0.009, 0.074]). Figure 6 presents the predicted scores on the IGD questionnaire as a function of BPNS and parental knowledge. The negative relationship between BPNS and IGD was significant among adolescents who reported lower levels of parental knowledge (1 *SD* below *M*; *β* = −0.13, *SE* = 0.03, *t* = −4.73, *p* < 0.01, 95% CI [−0.184, −0.076]) and non-significant among those who reported higher levels of parental knowledge (1 *SD* above *M*; *β* = −0.05, *SE* = 0.03, *t* = −1.86, *p > 0*.05, 95% CI [−0.096, 0.002]). Therefore, the relationship between BPNS and IGD was moderated by parental knowledge.

The bias-corrected percentile bootstrapping method was used to examine the conditional indirect effects of peer victimization on IGD as a function of parental knowledge. Specifically, the indirect relationship between peer victimization and IGD via BPNS was significant for adolescents who reported lower levels of parental knowledge (indirect effect = 0.03, *SE* = 0.01, 95% CI [0.014, 0.042]) and non-significant for those who reported higher levels of parental knowledge (indirect effect = 0.01, *SE* = 0.01, 95% CI [−0.002, 0.003]). Therefore, the moderating effect of parental knowledge on the relationship between peer victimization and IGD among adolescents was mediated by BPNS.

## 4. Discussion

This study tested a mediated moderation model based on the integration of the risk-buffering model [20,21] and self-determination theory [22,23]. First, we observed that the positive association between peer victimization and adolescent IGD was stronger among adolescents who reported having parents with low parental knowledge than those reporting high parental knowledge. This finding supports Hypothesis 1 and demonstrated how the influence of peer victimization on adolescent IGD can differ based on the degree of parental knowledge about their adolescents’ daily activities. Moreover, this result confirms the findings from previous research [24,33]. The risk-buffering model [20,21] contends that positive family factors robustly buffer the negative effects of a risky environment on adolescent development. Parents with greater knowledge about their adolescents’ whereabouts, activities, and companions appear to be able to predict the consequences of adolescents’ experience of peer victimization [29]. Therefore, parents may be able to attenuate the risk of IGD among adolescents experiencing peer victimization by providing timely assistance, such as stopping the victimization. Moreover, a high level of parental knowledge included more effective guidance and control [27], contributing to adolescents’ ability to regulate their behavior. Therefore, adolescents experiencing peer victimization whose parents have high parental knowledge are more likely to limit their use of online games compared to adolescents whose parents have low parental knowledge, consistent with the risk-buffering model. Adequate parental knowledge serves as a protective factor buffering against the adverse influence of peer victimization on adolescent IGD. These findings confirm the interaction of familial and peer influences (i.e., parental knowledge and peer victimization) in predicting adolescent IGD. Additionally, they provide further evidence for the value of the risk-buffering model in understanding the development of adolescent IGD.

Second, the results also demonstrated that the moderating effect of parental knowledge on the relationship between peer victimization and adolescent IGD was mediated by BPNS. Specifically, the negative association between peer victimization and BPNS was stronger among adolescents who reported having parents with high parental knowledge. This finding indicates that despite high parental knowledge playing a protective role in the relationship between low levels of peer victimization and BPNS, its benefits were not observed under high levels of peer victimization. The view of “cost of resilience” [61] provides a possible explanation for this finding. Individuals might pay the price for positive development within a high-risk environment [61]. This “price” is that protective factors can buffer the negative effect of adversity in one field, but increase the impact of adversity in other fields [61,62]. In other words, parental knowledge buffered the positive association between peer victimization and IGD, but the “cost” was the contribution to the adverse effect of peer victimization on BPNS. Previous studies have reported similar findings [62,63]. It has been suggested that under a high-risk environment, especially the influence of this high-risk environment is more significant than that of protective factors, and a single protective factor (i.e., parental knowledge) cannot counteract the deleterious outcome (i.e., the occurrence of IGD and the failure to meet basic psychological needs) [64]. Therefore, in order to promote adolescents’ positive development, it is necessary to intervene in terms of both environmental factors and protective factors.

Third, we observed that parental knowledge moderated the link between BPNS and adolescent IGD. Specifically, the negative association between BPNS and IGD was significant among adolescents with parents having low parental knowledge, but not significant for those with high parental knowledge. This finding demonstrates that high parental knowledge with low BPNS was also conducive to reducing adolescents’ risk of developing IGD. This result is similar to the findings of Ang et al. [43] According to self-determination theory [22,23], deficits in BPNS may lead to individuals neglecting social norms, as they are eager to meet their psychological needs. Particularly in the context of low parental knowledge, the attachment and emotional bond of adolescents to their parents would most likely also be low. Lacking positive parental guidance, adolescents may regard online games as a way to fill the void and seek psychological compensation, which could potentially result in IGD. Importantly, why was the link between BPNS and IGD not significant among adolescents with high parental knowledge? A possible explanation for this is that adequate parental knowledge is an important source that helps satisfy adolescents’ basic psychological needs and prevents the development of IGD. Adolescents with high parental knowledge may receive psychological compensating and satisfaction through having a positive parent–adolescent relationship. Therefore, they would be less likely to seek psychological compensation through the use of addictive online games.

Finally, it is meaningful to discuss our results from a Chinese cultural perspective. Influenced by Confucian culture for over 2000 years, Chinese people are accustomed to implicitly expressing their sentiments. Chinese parents seldom express their love directly to adolescents. They usually focus on and actively ask about the adolescents’ daily experiences as expressions of concern and love. For Chinese parents, parental knowledge may be an effective way to buffer peer victimization, satisfy basic psychological needs, and reduce the risk of IGD among adolescents. In addition, Chinese adolescents who are deeply influenced by the Confucian cultural norm of “filial piety” will be more obedient to their parents’ guidance and willing to accept their help [65]. Therefore, aspects of Chinese culture may be a factor that influences whether adolescents experience a lack of basic psychological needs satisfaction and IGD.

## 5. Limitations and Future Directions

Several limitations need to be acknowledged regarding the present study. First, the cross-sectional design is widely used in the field of IGD [33,37]. However, it is not possible to determine the causal relationships and dynamic information in the bidirectional relationship between peer victimization and IGD. Therefore, longitudinal studies should be used to replicate and further examine our current results. Second, the data were collected using adolescent self-reports, which can be affected by shared-method bias. Additional research should be conducted utilizing multiple informants and methods of data collection (e.g., parent assessment and peer nomination). Third, the participants in this study were from southern China. Considering that China is a vast country with a large multiethnic population, future research should verify our findings among other cultures, groups, and regions. Fourth, this study only considered the protective effect of parental knowledge on the association between peer victimization and IGD. Future studies should examine the protective role of schools, peers, and even other family factors/characteristics to identify a more comprehensive buffering mechanism in the association between peer victimization and IGD. Finally, the current study used questionnaires to explore the impact factor and mechanisms of IGD; however, it did not conduct any inquiries from cognitive and neural perspectives. Previous studies have revealed the cognitive function and neural mechanism of IGD through event-related potential (ERP) and functional magnetic resonance imaging (fMRI) [66,67]. Therefore, future studies should use multiple measurement methods to further investigate the brain functions related to IGD.

## 6. Implications for Practice

Despite having several limitations, this study has some valuable practical implications. First, the present study once again confirmed that peer victimization is a risk factor for adolescents in terms of developing IGD [17,18,19]. It is recommended that educators and parents are made aware of the existence of peer victimization among adolescents in a timely manner and devote more attention to encouraging them to build harmonious peer relationships. Second, this study revealed that parental knowledge has a robust buffering effect against the adverse influences of peer victimization on adolescent IGD. It is recommended that parents increase their knowledge of their adolescent’s life through positive communication, building intimate parent–adolescent relationships, and providing adolescents with adequate warmth and care. Third, our results revealed that BPNS mediates the moderating influence of parental knowledge. As such, addressing BPNS among adolescents who experience peer victimization could be an effective strategy for decreasing IGD. Therefore, teachers, parents, and practitioners should identify and focus on adolescents whose BPNS is thwarted. Furthermore, interventions can be developed based on the content of the three basic psychological needs (e.g., developing abilities, improving interpersonal relationships, and providing space for autonomy). Finally, adolescents should realize that virtual online games are not an appropriate method for meeting their psychological needs and escaping peer victimization. Adolescents need to learn appropriate methods for coping with peer victimization (e.g., seeking help from parents and teachers, and self-regulation), thereby protecting them from developing IGD.

## 7. Conclusions

The results of this study replicate and extend previous studies. It showed that parental knowledge should be considered as a protective role to buffer the risky effect of peer victimization on IGD among adolescents. Moreover, this moderating effect was mediated via BPNS. These findings promote a deeper understanding of the impact factors and mechanisms in adolescent IGD, and provide a meaningful implication for the prevention and intervention of IGD among adolescents.

## Figures and Tables

**Figure 1 ijerph-18-02397-f001:**
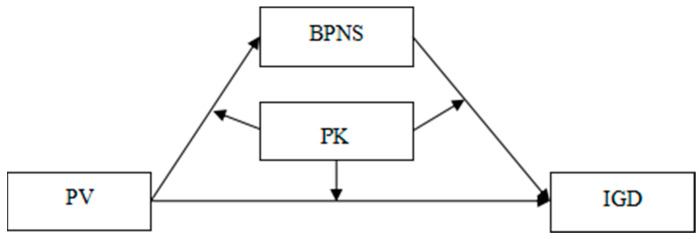
The proposed mediated moderation model. Note: PV = peer victimization, PK = parental knowledge, BPNS = basic psychological needs satisfaction, and IGD = Internet gaming disorder.

**Figure 2 ijerph-18-02397-f002:**
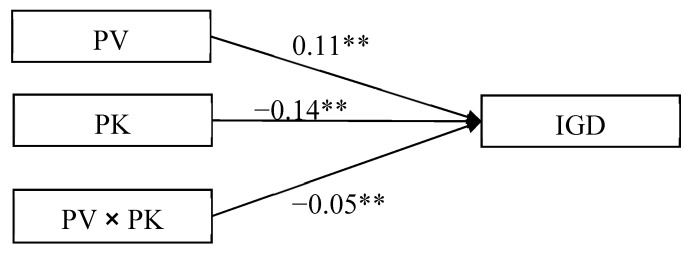
Model of the moderating role of parental knowledge between peer victimization and IGD. Note: PV = peer victimization, PK = parental knowledge, and IGD = internet gaming disorder. Values are standardized coefficients. ** *p* < 0.01.

**Figure 3 ijerph-18-02397-f003:**
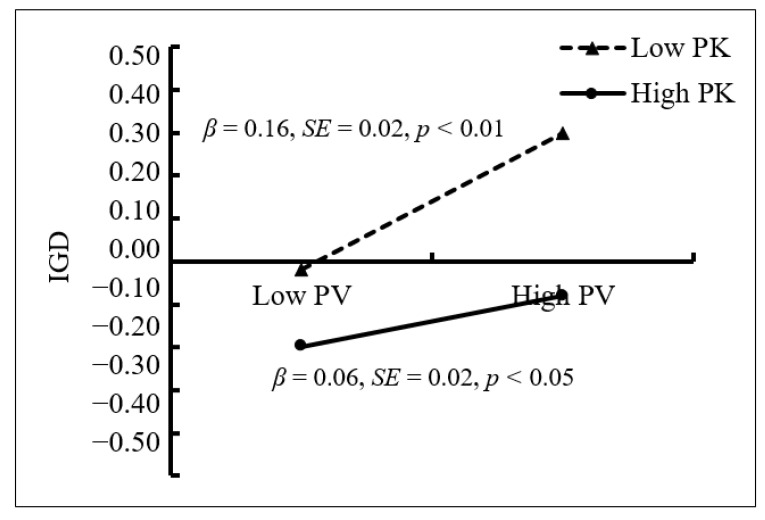
IGD among adolescents as a function of peer victimization and parental knowledge. Note: PV = peer victimization, PK = parental knowledge, and IGD = internet gaming disorder.

**Figure 4 ijerph-18-02397-f004:**
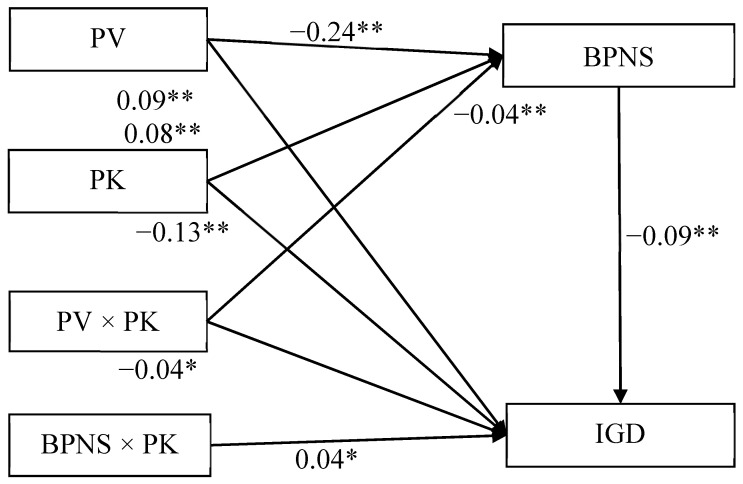
Model of the moderating role of parental knowledge in the indirect relationship between peer victimization and IGD. Note: PV = peer victimization, PK = parental knowledge, BPNS = basic psychological needs satisfaction, and IGD = internet gaming disorder. Values are standardized coefficients. * *p* < 0.05, and ** *p* < 0.01.

**Figure 5 ijerph-18-02397-f005:**
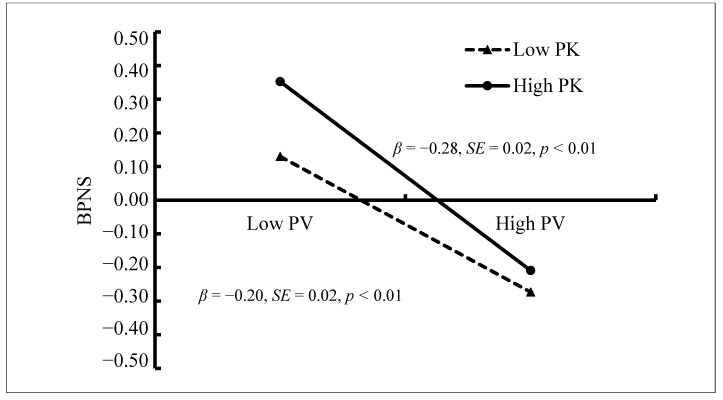
Basic psychological needs satisfaction among adolescents as a function of peer victimization and parental knowledge. Note: PV = peer victimization, PK = parental knowledge, and BPNS = basic psychological needs satisfaction.

**Figure 6 ijerph-18-02397-f006:**
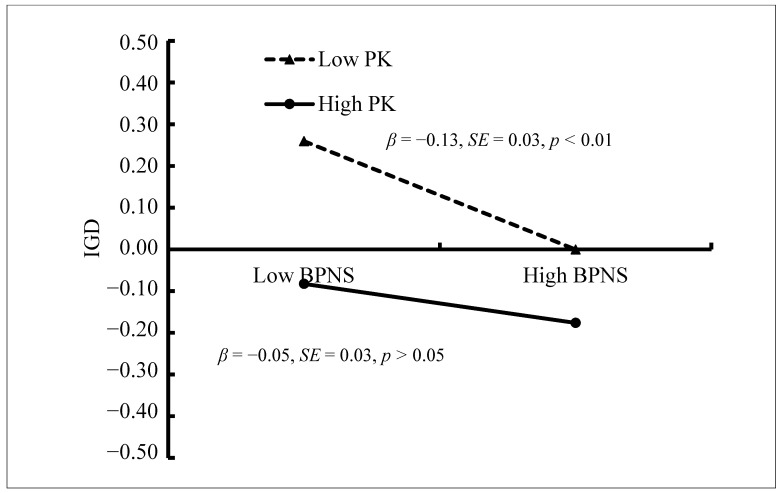
IGD among adolescents as a function of basic psychological needs satisfaction and parental knowledge. Note: PK = parental knowledge, BPNS = basic psychological needs satisfaction, and IGD = internet gaming disorder.

**Table 1 ijerph-18-02397-t001:** Descriptive statistics and correlations for all variables.

Variables	1	2	3	4	5	6	7	8
1. Gender	1.00							
2. Age	−0.02	1.00						
3. Impulsivity	−0.01	0.19 **	1.00					
4. PAR	0.03	−0.13 **	−0.33 **	1.00				
5. PV	0.07 **	−0.01	0.26 **	−0.22 **	1.00			
6. PK	−0.13 **	−0.28 **	−0.31 **	0.32 **	−0.14 **	1.00		
7. BPNS	0.06 **	−0.21 **	−0.50 **	0.42 **	−0.37 **	0.29 **	1.00	
8. IGD	0.33 **	−0.00	0.26 **	−0.19 **	0.22 **	−0.25 **	−0.22 **	1.00
Mean	0.47	14.93	2.29	2.40	1.72	2.37	3.49	1.40
*SD*	0.50	1.96	0.37	0.32	0.89	0.49	0.57	1.60

Note: Gender and age were dummy coded such that 1 = male and 0 = female. PAR = parent-adolescent relationship, PV = peer victimization, PK = parental knowledge, BPNS = basic psychological needs satisfaction, and IGD = internet gaming disorder. ** *p* < 0.01.

## Data Availability

The data presented in this study are available on request from the corresponding author (C.Y.).

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
