# Peer review of "The Influence of Parental Knowledge and Basic Psychological Needs Satisfaction on Peer Victimization and Internet Gaming Disorder among Chinese Adolescents: A Mediated Moderation Model"

_ijerph, 2021, doi:10.3390/ijerph18052397_

Round 1
Reviewer 1 Report
This paper is very interesting and valuable but we must come accept that peer victimization questionnaireInternet gaming disorder questionnaire and the basic psychological needs questionnaire are completely scientifically inadequate . for proper study of the needs for society and the quest understand the science of behavior each adolescent requires millon axis one / two instruments type and temperament i.e. Myers-Briggs neuropsychological assessment such as TOVA or CnsVS and some Neurological assessment such as P 300 and Q eeg / spect . other social factors as well a Framingham type national study of adolescent behavior needs to be far more comprehensive in the future. Still please accept the paper with a discussion of some of these caveats
Reviewer 2 Report
Dear authors,
I congratulate you on the work done. The subject of the article is of great relevance and brings new conclusions to the knowledge of victimization among peers and Internet gambling disorder.
I only have a few suggestions that would be interesting to apply.
- In my opinion, I think the title does not reflect the importance of the study. Perhaps it would be better if the title was changed to something like "The Role of Parental Knowledge and Satisfaction of Basic Psychological Needs in the Relationship between Peer Victimization and Internet Gaming Disorder in Chinese Adolescents."
- PK does not appear in figure 1, only P. In turn, the text that precedes figure 1 ("... moderated the relationship between victimization among peers and IGD and (2) if BPN.me-gave this effect moderator. Figure 1 illustrates the proposed research model. ") Add S to BPN.
- The rest of the figures show information that is repeated in the results. I suggest that this information be removed from the figures, as you can refer to the wording of the results. In Figures 2 and 4, you would keep the indices of b but suppress the values ​​of SE.
I am confident that you will be able to make these changes without any problems.
Good job!!
Reviewer 3 Report
Title: The influence mechanism of peer victimization and parental knowledge on adolescent internet gaming disorder: from the perspective of the self-determination theory
This cross-sectional survey of Chinese middle school adolescents examined whether parental knowledge moderated the relationship between peer victimization and adolescent internet gaming disorder and whether this moderating effect was mediated by basic psychological needs satisfaction. The study provided important evidence of the risk-buffering effects of high parental knowledge except under high-risk environments such high levels of peer victimization and low basic psychological needs satisfaction then the effects of this positive factor were attenuated. As a result, the authors were able to characterize possible practice implications that may arise from these findings. Although, the study appeared well conceived, the following issues need to be addressed:
- The authors needed to explain the following methodological issues:
- Why were 10-year-old children included in a study of adolescents?
- The authors should have commented on the validity of the IGD questionnaire in identifying adolescents with internet gaming disorder.
- Why are impulsivity and parent-adolescent relationship included as covariates. The concept of parental knowledge appears to include aspects of parent-adolescent relationship so why is it included as a covariate?
- The authors should have indicated the extent of missing data in their study measures.
Minor issues:
- In figure 1, the box noting “parental knowledge” should have the abbreviation “PK”.
- In section 3.2, the authors used the term “parental monitoring”; however, this term is not used anywhere else in the paper. Should they be referring to “parental knowledge”?
- In figure 3, there is no results related to BPNS so this should be removed from the Figure notes.
Reviewer 4 Report
The manuscript contains should results that might have interesting results with practical implications. The manuscript is concise, clear and well written. But authors must overcome serious issues before being considered for publication. These are my comments:
The term large in sentence …by a large body of empirical research… is superfluous. Please, authors should consider is it is adequate to use this term.
It should be necessary to add a couple of protective factors as examples before introducing the parental knowledge, as it seems to be several of them but only one is explained in depth.
The risk-buffering model and self-determination theory should be succinctly described as many of the readers of the manuscript will be not familiar with them.
Also, a short section on gaps in the literature and how the current study will fill in those gaps would be beneficial. This section should be included before study aims/hypotheses.
In figure one there’s not a PK acronym, just appears a “P” which is assumed to be PK. Please, if it is a mistake, correct.
A paragraph is required where data preparation is outlined. How were the missing values handled? Was an imputation method included?
Authors should comment on the validity of the scales used in the current study. Are they psychometrically established scales? References to reliability and validity studies performed on those scales should be included.
Which kind of school/high schools was data collected? (public, private, etc.)
Author should disclose why they chose the range of 10 -20 as adolescents when the UN defines it as between 10 and 19, Therefore 20 is above and out of adolescent and it is considered youth.
Regarding the measures, is there any reason for not stating that the scales are in Likert format.
Round 2
Reviewer 4 Report
Authors have addressed my comments and concerns satisfactorily